# Mad2 Induced Aneuploidy Contributes to *Eml4-Alk* Driven Lung Cancer by Generating an Immunosuppressive Environment

**DOI:** 10.3390/cancers13236027

**Published:** 2021-11-30

**Authors:** Kristina Alikhanyan, Yuanyuan Chen, Kalman Somogyi, Simone Kraut, Rocio Sotillo

**Affiliations:** 1Division of Molecular Thoracic Oncology, German Cancer Research Center (DKFZ), Im Neuenheimer Feld 280, 69120 Heidelberg, Germany; k.alikhanyan@dkfz-heidelberg.de (K.A.); sh.chenyy@gmail.com (Y.C.); k.somogyi@dkfz-heidelberg.de (K.S.); simone.kraut@gmx.de (S.K.); 2Translational Lung Research Center Heidelberg (TRLC), German Center for Lung Research (DZL), 69120 Heidelberg, Germany

**Keywords:** lung cancer, aneuploidy, tumor microenvironment, immune suppression, mouse models

## Abstract

**Simple Summary:**

In tissue homeostasis, aneuploid cells have been suggested to be recognized and eliminated by immune cells. However, these initiating cancer cells can evade the immune system and ultimately derivate a tumor. To better understand how aneuploidy might contribute to tumor initiation and/or progression, we used two lung cancer models: one in which cancer cells are aneuploid and the surrounding normal epithelial cells are diploid, and a second one in which both tumor and normal cells are aneuploid. We show that aneuploid cells surrounding the tumor generate an immunosuppressive environment that contributes to lung cancer initiation.

**Abstract:**

Aneuploidy, an imbalance number of chromosomes, is frequently observed in lung cancer and inversely correlates with patient survival. Paradoxically, an aneuploid karyotype has detrimental consequences on cellular fitness, and it has been proposed that aneuploid cells, at least in vitro, generate signals for their own elimination by NK cells. However, how aneuploidy affects tumor progression as well as the interplay between aneuploid tumor cells and the tumor microenvironment is still unclear. We generated a new mouse model in which overexpression of *Mad2* was almost entirely restricted to normal epithelial cells of the lung, and combined it with an oncogenic *Eml4-Alk* chromosome inversion. This combination resulted in a higher tumor burden and an increased number of tumor nodules compared to control *Eml4-Alk* mice alone. The FISH analysis detected significant differences in the aneuploidy levels in the non-tumor regions of *Eml4-Alk*+*Mad2* compared to *Eml4-Alk* alone, although both tumor groups presented similar levels of aneuploidy. We further show that aneuploid cells in the non-tumor areas adjacent to lung tumors recruit immune cells, such as tumor-associated macrophages. In fact, these areas presented an increase in alveolar macrophages, neutrophils, decreased cytotoxic CD8^+^ T cells, and IFN-γ, suggesting that aneuploid cells in the surrounding tumor areas create an immunosuppressive signature that might contribute to lung tumor initiation and progression.

## 1. Introduction

Lung cancer (LC) is the leading cause of cancer-related deaths worldwide, and the 5-year relative survival is one of the lowest (21%) [1]. Non-small cell lung cancer (NSCLC) accounts for approximately 85% of all LC cases. Among those, the rearrangement between the Anaplastic lymphoma kinase (ALK) and the echinoderm microtubule-associated protein-like 4 (*EML4*) is identified in 3–7% of NSCLC cases, most commonly in younger patients, non-smokers, and those with adenocarcinoma histology [2].

Aneuploidy, defined as the presence of an incorrect number of chromosomes, is one of the most common characteristics of human cancers [3]. High-level deviation from a diploid karyotype has been described in 64% of NSCLC, and patients with aneuploid tumors present a shorter survival than patients with nearly diploid tumors [4]. An improved understanding of the molecular mechanisms of aneuploidy has provided important insights into the complex relationship between chromosome number alterations and cancer. Evidence obtained in nonmalignant cells and cancer models indicates that chromosome instability can favor the development and selection of malignant clones, driving progression to a very aggressive phenotype [5]. On the contrary, aneuploidy can also inhibit tumor growth [6,7] or reduce the tumorigenicity of cells harboring extra chromosomes [8]. Whether aneuploidy will drive or inhibit tumorigenesis might be context-dependent [9]. In addition, it has been shown that the degree of tumor aneuploidy correlates with markers of immune evasion as well as with the reduced response to immunotherapy [10,11]. However, other studies suggest that aneuploidy is associated with the activation of some immune responses. For instance, a recent study from Santaguida and colleagues [12] found that NF-kB upregulation is critical for NK cell-mediated clearance of untransformed aneuploid cells. Nevertheless, additional mechanisms in cancer cells may counteract NF-kB mediated immunogenicity and render cancer cells insensitive for NK cell-mediated killing. Although genomic analyses of human data and cell culture models have generated some new insights into this complex relationship between aneuploidy and the immune system, some of these approaches are still limited as they do not fully recapitulate tumors in patients as they lack the appropriate in vivo environment. 

The link between aneuploid cancer and the immune system is slowly beginning to be uncovered. Nevertheless, the impact that an aneuploid tumor environment might have on tumor progression has not yet been studied. It has been shown that aneuploid NSCLCs are frequently associated with aneuploid preinvasive lesions [13], but the influence that this aneuploid environment might have on the immune response is lacking. 

We previously reported that aneuploidy, driven by overexpression of the spindle assembly checkpoint protein Mad2, accelerates lung tumor progression when combined with the expression of *Kras^G12D^* [14] and contributes to the development of persistent subclones in breast cancer models, that continue to grow even after oncogene withdrawal, thereby facilitating tumor relapse [6,15].

Based on these models, here we investigated the biological interplay between an aneuploid environment and LC to understand the effects of aneuploidy in the progression of NSCLC. We demonstrate that aneuploidy caused by Mad2 overexpression promotes the tumorigenesis of *Eml4-Alk* lung adenocarcinoma development. Moreover, our findings reveal that an aneuploidy environment increases the number of alveolar macrophages (AMs) as well as tumor-associated macrophages (TAMs) and neutrophils, while the level of cytotoxic CD8^+^ T cells and IFN-γ in lung tissue decreases, thus generating an immunosuppressive profile, that could lead to a higher tumor burden.

## 2. Materials & Methods

### 2.1. Experimental Model

Mice with a FVB background were bred at DKFZ animal facilities where the health status was constantly monitored by the veterinary staff. Animal care and experimentation were performed in accordance with guidelines of EU animal laws and had been approved by the Animal Welfare and Ethical Review Bodies of Baden-Wurttemberg, Germany (animal license No. G185-17). *Eml4-Alk* driven lung adenocarcinoma in mice [14] was induced by intratracheal instillation of *Eml4-Alk* adenovirus in 6-week-old mice [16] (Alk group). Prior to viral administration, mice were anesthetized by intraperitoneal injection of 100 μg g^−1^ ketamine and 14 μg g^−1^ xylazine. To induce aneuploidy in the lung, *KH2-Mad2* [6] mice were crossed with *CCSP-rtTA* mice [14] referred to as (Mad2 group) and fed with doxycycline, administrated via impregnated food pellets (625 mg/kg: Harlan-Teklad) for transgene induction at 6 weeks of age. These tetracycline-inducible animals were generated using KH2 ES cells (Thermo Scientific, Waltham, MA, USA) according to a previously described method [17]. Murine *Mad2* cDNA was amplified with specific primers containing the HA epitope tag and ligated into the EcoRI site of Flp-in vector pBS31′. Electroporation of pBS31′-mHA-Mad2 together with pCAGGS-FLPe vector into KH2 ES cells resulted in the targeted integration into the ColA1 locus. KH2 ES positive clones expressing HA-Mad2 were injected into 8 cell blastocysts. Four chimeras from two independent ColA1-HA-Mad2 clones achieved germline transmission. Animals were backcrossed to FVB. To exclude the Rosa26-M2rtTA from the KH2 ES cells crosses with the *CCSP-rtTA* line followed. *KH2-Mad2/CCSP-rtTA* mice were transduced with *Eml4-Alk* adenovirus and at the same time fed with doxycycline food to generate Alk+Mad2 group. Mice were randomly assigned to different experiments, and investigators were not blinded with respect to which animal *Eml4-Alk* adenovirus was injected. Mice at the age of 12-weeks (6 weeks of treatment) or 22-weeks (16 weeks of treatment) were killed with a ketamine/xylazine overdose, and the lungs were perfused with 10 mL PBS through the right ventricle.

### 2.2. Fluorescence In Situ Hybridization

Formalin-fixed paraffin-embedded 5-µm sections of mouse lung tissue were hybridized with probes for chromosomes 12, 16, and 17, as described previously [18]. Briefly, after deparaffinization, rehydration, and antigen retrieval with unmasking solution 0.09% (*v/v*) (Vector Labs, California, USA, NA), hybridization was performed using the Abbott Molecular Thermobrite system with the following program: denaturation 76 °C for 5 min, hybridization at 37 °C for 20–24 h. Pan-centromeric probes were made using pairs of BAC clones for each chromosome. Chr 12-RP23-54G4 and RP23-41E22 labeled with SpectrumRed-dUTP (Abbott GmbH, Wiesbaden, Germany), Chr 16-RP23–290E4 and RP23–356A24 labeled with SpectrumOrange-dUTP (Vysis), and Chr 17-RP23–354J18 and RP23–202G20 labeled with SpectrumGreen-dUTP (Vysis); the BAC DNAs were labeled by nick translation according to standard procedures. Probe mix was prepared with 3 µL of the labeled probe and 7 µL Vysis LSI/WCP Hybridization buffer. Post-hybridization washes were performed as described in [15].

FISH samples were imaged with a Zeiss Cell Observer microscope (Oberkochen, Germany) (×63, NA 1.3 oil) at the DKFZ Light Microscopy Facility. Multiple focal planes were acquired for each channel to ensure that signals on different focal planes were included. The resulting fluorescence emissions were collected using 425–475 nm (for DAPI), 546–600 nm (for spectrum orange), 650–670 nm (for spectrum far red), and 500–550 nm (for AlexaFluor488) filters. Signal for hybridization for each probe was checked in a minimum of 100 interphase cells with strong and well-delineated contours in each sample (in both tumor and non-tumor areas). Images were analyzed using FIJI software.

### 2.3. Bronchoalveolar Lavage

Under anesthesia with ketamine/xylazine, a catheter was inserted into the trachea and lungs were lavaged with 1 mL ice-cold 0.6 mM ethylenediaminetetraacetic acid in PBS. The lavage was performed eight times, and the collected bronchoalveolar lavage fluid (BALF) was centrifuged for 10 min at 2000 rpm. The resulting supernatant was subjected to IL-6 measurement using mouse IL-6 sandwich ELISA (R&D, Mouse IL-6 DuoSet ELISA) according to the manufacturer’s protocol. The cell pellet was resuspended in 1 mL of PBS, and the total number of living cells in the BALF was counted with a Cellometer using trypan blue staining. A hundred cells were analyzed by fluorescence-activated cell sorting (FACS) for AMs, and the percentage was extrapolated to a total number of AMs in the BALF.

### 2.4. Histology

Following euthanasia and lung perfusion with PBS, whole lungs were excised and weighed. The left lung was used for tissue homogenization and FACS analyses, and the right lung was fixed with 10% formalin (Sigma-Aldrich, Taufkirchen, Germany, HT501128) for further histological analyses. After processing in a tissue processor (Leica, Mannheim, Germany ASP300S), the tissue was embedded in paraffin and cut into 5 μm sections. Hematoxylin and eosin staining was performed to assess the lung pathological changes using light microscopy. The number of nodules, as well as the area of each nodule, was quantified using the StrataQuest Analysis Software (Tissuegnostic, Vienna, Austria). 

For immunofluorescent staining, the following primary antibodies were used: anti-HA (1:200, Roche, Basel, Switzerland 11867423001), anti-CCSP (1:1000, Merck-Millipore, Massachusetts, USA, NA 07-623), anti-SPC (1:500, Merck-Millipore, AB3786). Secondary antibodies were Alexa 488 donkey anti-rat IgG (1:500, ThermoFisher Scientific, A21208), Alexa 568 donkey anti-rabbit IgG (1:500, ThermoFisher Scientific, Massachusetts, USA, NA A10042). Pictures were taken in a Leica SP5 confocal system and Tissuegnostic TissueFAX system, Vienna, Austria.

Sections for immunohistochemistry were deparaffinized and rehydrated using xylene and graded ethanol solutions. A citric acid solution was used for antigen retrieval at 94 °C using a steamer. Tissue sections were then incubated in 3% hydrogen peroxide for 15 min to block endogenous peroxidase activity. Slides were blocked with 10% normal goat serum for 1 h before application of antibodies against CD8 T cells (1:100; 14-0808-80; ThermoFisher Scientific), Ki67 (ready-to-use; 275R-18; Medac diagnostika, Wedel, Germany), PCNA (1:1000, Abcam, Cambridge, United Kingdom, ab18197), and HA (1:200, Roche, 11867423001). Sections were then subjected to sequential incubations with the indicated biotinylated secondary antibodies and avidin/biotinylated enzyme reagents (ABC kit (PK-6101; PK-61-04)). For visualization, the DAB peroxidase substrate kit (SK-4100) was used according to the manufacturer’s instructions. The sections were counterstained with Harris’ modified hematoxylin (Sigma-Aldrich, Taufkirchen, Germany, HHS16).

Multiplexed immunohistochemical consecutive staining on a single slide [19] was used to determine HA (1:200, Roche, 11867423001), Iba1 (1:200, Fujifilm Wako, Neuss, Germany, 019-19741), and CD206 (R&D Systems; AF2535, 1:500) positive cells in the same tumor and non-tumor regions of the lung. In this case, a serum-free protein block solution (Dako, Glostrup, Denmark X0909) for 30 min was used to block free Fc receptor binding sites before adding the primary antibodies, followed by biotinylated secondary Abs. The binding of biotinylated Abs was revealed using streptavidin-HRP, and chromogenic revelation was performed using AEC (Vector, SK-4200). Nonspecific isotype controls were used as negative controls. Tissue sections were then counterstained with hematoxylin, mounted with aqueous mounting medium (Dako, C0563), and scanned for digital imaging and quantification (TG3-951I TissueFAXS-i-plus system for the scanning and analysis of slides with StrataQuest Analysis Software). After scanning, slide coverslips were removed in hot (~50 °C) water, and tissue sections were destained in organic solvent (50% ethanol, 2 min; 100% ethanol, 2 min; 100% xylene; 2 min, 100% ethanol, 2 min; and 50% ethanol, 2 min). Then, the slides were directly subjected to the next round of staining as previously described with some modifications. Antigen retrieval was performed before incubating each slide with 3% hydrogen peroxide plus 1 mM sodium azide for 20 min and in serum-free protein block solution (Dako) for 30 min. Slides were then blocked with 5% serum and monovalent Fab fragment for 30 min. Serum (Jackson ImmunoResearch, Ely, United kingdom) was from the same species as the previous primary Ab used, and Fab fragment (Jackson ImmunoResearch) was directed against the same species as the primary Ab used previously. Then, the tissue sections were stained as previously described.

Each stain was artificially attributed to a color code, and images were overlaid using ImageJ. Ten regions of interest of 1 mm^2^ each per section were analyzed with StrataQuest Analysis Software.

### 2.5. Determination of IFN-γ Concentration

Cytokine analysis was conducted using a mouse Cytokine Array Panel A (ARY006, R&D systems, Inc., Minneapolis, MN, USA) according to the manufacturer’s protocol. Briefly, tumor and non-tumor parts of lung tissue homogenates were diluted and mixed with a cocktail of biotinylated detection antibodies. The sample/antibody mixture was then incubated with the Mouse Cytokine Array membrane containing 40 different anti-cytokine antibodies printed in duplicates. Any cytokine/detection antibody complex present was bound by its cognate immobilized capture antibody on the membrane. Following a wash to remove unbound material, Streptavidin-HRP and chemiluminescent detection reagents were added sequentially. The membrane was exposed to an X-ray film for 1–10 min. Light was produced at each spot in proportion to the amount of cytokine bound. The average signal (pixel density) of the pair of duplicate spots representing each cytokine was analyzed by ImageJ.

### 2.6. FACS Analyses

The tumor and non-tumor areas from the left lung of the mouse were separately dissociated into single-cell suspensions using the gentleMACS Dissociator. Red blood cell lysing buffer (Sigma–Aldrich, Taufkirchen, Germany) was used for red cell lysis. For the isolation of viable lymphocytes from murine lungs, single-cell suspension Lympholyte-M (Biozol Diagnostica, Eching, Germany, CED-CL5035), a density separation medium, was used. Cells were then blocked with anti-mouse CD16/32 (101319, Biolegend, San Diego, CA, USA) and then stained with antibodies Brilliant Violet 421 anti-mouse CD11b (101235, Biolegend, San Diego, CA, USA), APC/Cy7 anti-mouse CD45 (103116, Biolegend, San Diego, CA, USA), PE/Cyanine7 anti-mouse CD64 (139313, Biolegend, San Diego, CA, USA), PerCP/Cy5.5 anti-mouse CD11c (117328, Biolegend, San Diego, CA, USA), PerCP/Cy5.5 anti-mouse CD4 (100540, Biolegend, San Diego, CA, USA), PE/Cy7 anti-mouse CD3ε (100320, Biolegend, San Diego, CA, USA), Brilliant Violet 421 anti-mouse CD335 (NKp46) (137612, Biolegend, San Diego, CA, USA), APC anti-mouse CD8a (100712, Biolegend, San Diego, CA, USA), and Zombie Aqua Fixable Viability Kit (423102, Biolegend, San Diego, CA, USA), FITC anti-mouse I-A/I-E (MHCII) (107605, Biolegend, San Diego, CA, USA), PE anti-mouse CD49a (562115, BD Biosciences, Franklin Lakes, NJ, USA), FITC anti-mouse CD49b (108905, Biolegend, San Diego, CA, USA), PE anti-mouse Siglec-F (552126; BD Biosciences, Franklin Lakes, NJ, USA), Brilliant Violet 650™ anti-mouse Ly-6G/Ly-6C (Gr-1) (108441, Biolegend, San Diego, CA, USA). In all FACS analyses, we first excluded cell debris, doublets, and dead cells. Single stains were performed for compensation controls, FMO controls, to check for fluorescence spread, and isotype controls, were used to determine the level of nonspecific binding.

Gating for CD45^+^AquaZombie^−^CD64^+^CD11c^−^CD11b^+^ cells was used for interstitial macrophages (IMs); CD45^+^AquaZombie^−^CD64^+^CD11b^+^MHCII^high/low^ to detect the M1 and M2 population of IMs CD45^+^AquaZombie^−^NKp46^+^ for NK cells, CD45^+^AquaZombie^−^CD3^+^CD8^+^ for CD8 T cells, CD45^+^AquaZombie^−^CD3^+^CD4^+^ for CD4 T cells, CD45^+^AquaZombie^−^SyglecF^+^CD11c^+^ for AM.

Flow cytometric data acquisition was performed on a BD FACS Canto II machine, and data analysis was performed using FlowJo software. 

### 2.7. Grouping Patients

The human lung adenocarcinoma (LUAD) single-cell dataset was published by Maynard et al. [20]. The CIN70 signature [21] was used to evaluate the level of CIN in the non-tumor epithelial cell population in the dataset. Briefly, the z-score was calculated in every single cell. The ones with a z-score of no less than 1 were considered to be high CIN, whereas the ones with a z-score of no higher than –1 were considered to be low CIN. For each patient, the number of high CIN and low CIN cells was counted, respectively. Patients with higher CIN cells than lower were grouped into “high”, otherwise, they were grouped into “low”.

### 2.8. Comparing T Cell and Macrophage Scores

We compared the patients’ T cell exhaustion, cytotoxic T cell, M1 macrophage, and M2 macrophage levels scored by the authors (T cell exhaustion and cytotoxic T cell) or by us (M1 and M2 macrophages; the gene signature for each was published [22], and the scores were generated by *AddModuleScore* from Seurat package). Then, violin plots of each signature were created according to each sample’s CIN group (high vs. low) among all the samples or those that were collected before treatment (treatment naïve).

### 2.9. Statistical Analyses

A Student’s *t*-test (two tailed) was used to assess the significance of differences between the two groups. In the case of comparing more than two groups, statistical significance was calculated by one-way analysis of variance (ANOVA). Data were analyzed using GraphPad Prism 8 (GraphPad software). The *p*-values  <  0.05 were considered statistically significant. The data shown in each figure represent the mean of three or more independent experiments. 

## 3. Results

### 3.1. Mad2 Overexpression Results in Increased Tumor Burden in Eml4-Alk Mice

To induce the *Eml4-Alk* chromosomal rearrangement in vivo, we infected animals intratracheally with an adenovirus vector that contained Cas9, a specific short, guided RNA (sgRNA) against *Eml4*, and a sgRNA for Alk [16]. We referred to this group as Alk mice. To induce aneuploidy in vivo, we infected mice that contained a doxycycline-regulated *HA-**Mad2* transgene under the *CCSP-rtTA* promoter [14] (hereafter Alk+Mad2). Upon doxycycline exposure, we found that *Mad2* was expressed in some Club cells in the airways and in Alveolar type II (AT2) cells in the distal lung, driving aneuploidy in these cell types (Figure 1A,B).

*Mad2* overexpression driven by *CCSP-rtTA* in a *Kras* mutant background causes aneuploidy in lung tumors [14]. Therefore, we sought to determine the aneuploidy levels in Alk and Alk+Mad2 tumors. FISH analysis using 3 chromosome-probes suggested that both tumor groups had almost similar aneuploidy levels (Figure 1C and Appendix A). We next stained the lung sections with an antibody against HA to determine whether *Mad2* was indeed expressed in these tumors. *Mad2* was not uniformly expressed in Alk+Mad2 tumors (Figure 1D and Appendix A) but present throughout the non-tumor areas of the lung, mainly in the AT2 cells of the normal distal lung. These results are in line with previous observations where *Mad2* was not equally expressed in breast tumors induced by mutant *Kras* [6]. Further quantification of the aneuploidy levels in the non-tumor areas showed that the percentage of cells with an abnormal number of chromosomes was 15–fold higher in the Alk+Mad2 group compared to Alk mice (Figure 1C).

To determine whether aneuploidy caused by *Mad2* overexpression accelerates tumor formation driven by *Eml4-Alk* oncogene, we assessed tumor burden in mice 16 weeks after adenoviral infection in control (Alk) and aneuploid groups (Alk+Mad2) measured by lung weight as well as by tumor area (Figure 1E,F). Both measurements showed a significantly higher tumor burden in Alk+Mad2 group compared to control. 

To better understand the increased tumor burden in the aneuploid group, we measured the proliferation of tumor cells using two different markers, PCNA and Ki67. Both markers revealed a similar proliferation rate in Alk and Alk+Mad2 tumors (Appendix A). Additional staining with Caspase3 suggested very low cell death, and no significant differences were observed between the two tumor genotypes. Since proliferation and apoptosis rates were similar between the two groups 16 weeks after tumor initiation, we measured the number and size of all visible nodules in the lungs of infected mice (Appendix A). Interestingly, the total number of nodules was increased in the aneuploid group (Figure 1G), while the size of the nodules was not higher (Appendix A), suggesting that the increased tumor burden in this group might be a consequence of the increased tumor initiation rather than tumor progression.

### 3.2. An Aneuploid Environment Recruits Tumor-Associated Macrophages

The tumor microenvironment has been recognized to be an essential determinant of tumor progression [23,24]. Moreover, tumor aneuploidy is a marker of immune evasion [11]. Therefore, we investigated the immune landscape in these tumors as well as in the non-tumor areas since the latter showed an increase in aneuploidy. We performed sequential immunohistochemical analyses with the macrophage markers Iba1 and CD206 to address the infiltration of tumor-associated macrophages (TAMs), which have been shown to be tumor-promoting. Analysis of tumor areas revealed no differences inside the tumors between control and aneuploid groups (Figure 2A–C), while the percentage of macrophages in the non-tumor area was significantly higher in the aneuploid group (Figure 2D–F). Interestingly, the percentage of CD206 positive macrophages, which represent the M2 subpopulation, in the non-tumor area of Alk+Mad2 was almost the same as in the tumors (Figure 2C,F). To validate these results, we performed FACS analyses from the isolated tumor nodules as well as the non-tumor areas. After the identification of immune cells with the pan-hematopoietic marker CD45, interstitial macrophages (IMs) were identified, based on their expression of CD64 and CD11b and the absence of CD11c (Figure 3A). These analyses confirmed no differences in the percentage of IMs in tumor areas between Alk and Alk+Mad2 groups. However, a significant increase in IMs in the non-tumor areas of the aneuploid group was observed (Figure 3B). 

We then analyzed CD11b^+^ cells for MHCII^high^ and MHCII^low^ subpopulations (Figure 3A) to further investigate the characteristics of macrophages within TAMs since it is known that M1 macrophages express high levels of MHCII [25,26]. As shown in Figure 3C, the percentage of M1-like macrophages having anti-tumor activity was significantly lower in the non-tumor area of Alk+Mad2 group. 

Alveolar and interstitial macrophages can both contribute to tumorigenesis [27,28]. However, in some cases, AMs can have a higher tumorigenic role compared to IMs, as we have recently shown in an EGFR-driven NSCLC model [29]. Therefore, we isolated and counted the number of AM in the bronquiolar alveolar lavage fluid (BALF). Indeed, we found a significant increase in AM in the aneuploid group compared to the control group (Appendix A). Additional flow cytometric analysis confirmed the purity of AM (Appendix A). Recently, it was reported that IL-6, which is produced by a variety of cells, including macrophages, has an important role in cancer initiation and progression [30,31]. Based on these data, we looked at the level of IL-6 in the BALF and found a significantly higher level of IL-6 in Alk+Mad2 group compared to Alk group (Appendix A), suggesting a potential role of IL-6 in the enhanced tumorigenesis.

### 3.3. The Aneuploid Environment Contains Reduced Number of CD8^+^ T Cells, an Increase in Neutrophils and Inactive NK Cells 

To further characterize the immune landscape of lung tissues, we looked at the presence of cytotoxic T cells by immunostaining with a CD8 antibody. No differences were found in the percentage of CD8^+^ T cells inside the tumors between Alk and Alk+Mad2 groups (Figure 4A and Appendix A). However, in non-tumor areas of Alk+Mad2 lungs, the percentage of cytotoxic T cells was significantly reduced compared to the Alk group (Figure 4B). Similar results were obtained by flow cytometry (Figure 4C–E). Moreover, the CD4 subpopulation of CD3 T cells, which represent a diverse cell population with many differentiation states, including CD4^+^ T_reg_ cells [32], was significantly higher in the non-tumor areas of Alk+Mad2 mice (Figure 4C–E). Altogether, the increase in CD4/CD8 ratio might result in increased tumorigenesis. 

Finally, we looked at the distribution of innate lymphocytes and, in particular, natural killer (NK) cells in lung tumors and non-tumor areas. Strikingly, the number of NK cells was higher in both the tumors and non-tumor areas of the aneuploid group (Appendix A). Gao and colleagues [33] showed that NK cells can lose their anti-tumor properties by conversion into an intermediate type 1 innate lymphoid cells (intILC1s) (CD45^+^NKp46^+^CD49a^+^CD49b^+^) and type 1 innate lymphoid cells (ILC1s) (CD45^+^NKp46^+^CD49a^+^CD49b^−^). In contrast to NK cells, which are known to have a crucial role in controlling tumor initiation and progression, intILC1s and ILC1s are unable to restrain tumor growth and metastasis. Thus, we hypothesized that the high number of NKp46+ cells in the presence of aneuploidy could be converting into ILC1 cells. However, flow cytometric analyses of CD45^+^NKp46^+^cells gated for the expression of CD49a and CD49b revealed a very low frequency of ILC1 and intILC1 in Alk and Alk+Mad2 groups (Appendix A), suggesting that in these lung cancer models, there is no conversion of NK cells into ILC1 nor intILC1.

We next performed a cytokine array and found that the level of IFN-γ in the aneuploid group was significantly lower compared to control in both tumors and non-tumor areas (Appendix A), suggesting a reduction in the cytolytic activity of NK cells despite their higher concentration, since IFN-γ is primarily secreted by activated T cells and NK cells [34].

We next examined neutrophils, characterized by high expression of Ly6G and known to be the most prevalent immune cell type in NSCLC [35]. Immunohistochemical analysis of Ly6G showed a higher percentage of neutrophils in the non-tumor area of Alk+Mad2 group compared to Alk and normal control lungs, but the tumors showed no differences between the groups (Appendix A). It is possible that the increased number of neutrophils may also play a role in tumorigenesis, as has been suggested previously [36]. This tumorigenic role can be achieved through multiple mechanisms, including DNA damage and gene mutation in premalignant epithelial cells, thus driving oncogenic transformation in lung cancer. 

### 3.4. Immune Landscape during Early Stages of Tumorigenesis

Altogether, our previous data supported a scenario in which an aneuploid environment recruits immune cells generating an immunosuppressive microenvironment. This contributes to the increased transformation of small clusters of cells, therefore increasing the overall tumor burden in Alk+Mad2 animals. However, these results are in contrast to published observations where aneuploid cells are eliminated by immune cells [37]. To better understand how the immune system was suppressed in our lung cancer model, we characterized an early stage of tumor initiation. We analyzed the tumor burden of mice 6 weeks after injection with *Eml4-Alk* adenovirus (Alk–6wks) and compared it to mice that, in addition, expressed Mad2 (Alk+Mad2–6wks). At this early time point, we failed to find significant differences in tumor burden (Figure 5A) and in the number of nodules (Figure 5B) between the two groups. We next quantified the number of AMs in the BALF. Since the tumor size at this time point was small, we also compared the results to animals that expressed Mad2 but were not infected with the *Eml4-Alk* virus. We found an increased number of AMs in the Alk+Mad2–6wks group, although not significant compared to the Alk group, while the number of AMs was clearly low in the Mad2 group since these mice did not develop tumors (Figure 5C). However, the percentage of IMs measured by FACS (Figure 5D), as well as the level of IL-6 in the BALF (Figure 5E), was significantly higher in the Mad2 group without *Eml4-Alk* compared to the other two groups. It is possible that at the initial stages when Mad2 is expressed, aneuploid cells exhibit a pro-inflammatory response, leading to an increase in macrophage infiltration, upregulation of inflammatory gene signatures, and an induction of MHC-I antigen presentation.

Immunohistochemical evaluation of T-cells, mainly infiltration of non-tumor areas of lung tissues by CD8^+^ cells, allowed us to reveal that their quantity was the lowest in the Mad2 group and the highest was seen in the Alk group. These data were also confirmed by flow cytometric analyses (Figure 5F). Furthermore, after 6 weeks, we also observed a significantly higher percentage of NKp46 cells in the Mad2 group, whereas no differences were seen between Alk and Alk+Mad2 groups (Figure 5G). 

### 3.5. Immune Landscape in Human Lung Adenocarcinoma Samples

We next explored the relevance of our findings in human lung adenocarcinoma (LUAD) samples to examine whether different levels of chromosome instability (CIN) in tumor-adjacent cells correlate with a different composition of the tumor immune microenvironment in patients. We made use of a single-cell transcriptomic dataset previously published that contained information of 49 biopsies of 30 LUAD patients from different stages of treatment: treatment naïve, residual disease, and progressive disease [20]. In this study, we mainly focused on the treatment of naïve patients that were not stratified based on driver mutation and therefore included ALK, EGFR, BRAF, ROS1, and KRAS tumors. Single cells were annotated as tumor cells, non-tumor epithelial cells, T cells, and macrophages. We first evaluated the level of CIN in non-tumor epithelial cells based on the CIN70 signature [21], which consists of a signature of chromosomal instability from specific genes whose expression is consistently correlated with total functional aneuploidy in several cancer types. For all the cells tested, we created a normal distribution of the z-scores of CIN70, and the two tails were classified as high CIN (no less than 1) and low CIN (no more than –1) cells. We next compared the number of high and low CIN cells in each patient and annotated those who had a higher number of CIN cells than lower CIN cells as high CIN patients. Patients with equal or lower CIN cells were annotated as low CIN patients. Based on this annotation, we then focused on T cells and macrophages of each patient (Figure 6A). We compared the levels of T cell exhaustion and cytotoxic T cells between high and low CIN patients according to the scores the authors calculated. Agreeing with our data obtained from the mouse model, we found that high CIN patients had a significantly higher level of T cell exhaustion signature but a lower level of cytotoxic T cell signature (Figure 6B). Moreover, we calculated the level of M1 and M2 macrophage signature within the macrophage population using the published gene lists [22]. We found a decrease in the M1 signature in the high CIN patients compared to low CIN patients, although no significant difference was observed in the M2 signature (Figure 6C). Altogether, our data analysis in human LUAD patients further supported our findings that a higher level of CIN in tumor-adjacent epithelial cells helps to create an immune-suppressive microenvironment that facilitates tumor initiation and growth.

## 4. Discussion

Aneuploidy is a common feature of tumors, but it is also found in normal as well as in aged tissues [38]. In addition, aneuploidy is frequently found in preneoplastic lesions of the peripheral parts of the lung [13]. However, the role of aneuploidy in adjacent surroundings of tumor cells is not well understood and characterized. Therefore, to better understand the role of aneuploidy in the surroundings of an initiating tumor, we made use of a mouse model in which overexpression of *Mad2* was almost restricted to normal epithelial cells in the lung. By combining this model with an oncogenic *Eml4-Alk* translocation that induces LUAD, we found no differences in aneuploidy in the tumors but an increase in the normal adjacent cells. 

The tumor burden in the animals that had an aneuploid environment was higher than in those with a diploid karyotype; results that were in line with studies underscoring the association of aneuploidy with a poor prognosis in many tumor settings [39]. Furthermore, we found that not the size, but the number of nodules determined the heavier tumor burden in Alk+Mad2 group, raising the possibility that an aneuploid environment could create fertile soil for the awakening of dormant cancer cells and tumor initiation. It is also possible that continuous overexpression of *Mad2* leads to the development of aneuploidy in normal tissue, thus affecting the normal immune microenvironment. Recent studies have disclosed the correlation between high levels of somatic copy number alterations and reduced immune activation in almost all cancer types [11]. By investigating the immune landscape of aneuploid lung tissues, we identified a higher infiltration of pro-tumorigenic M2 macrophages compared to the euploid counterparts. This higher number of IMs was already detectable at earlier time points in mice that only overexpressed Mad2, suggesting that aneuploidy in normal epithelial cells of the lung recruits anti-inflammatory macrophages. In addition, a stronger reduction in M1-like macrophages was evident in the surrounding tumor areas that were highly aneuploid, compared to the euploid group. Remarkably, these changes correlate with findings in human LUAD, where patients with high CIN scores presented lower levels of M1 macrophages. Although the mechanism by which aneuploidy alters macrophages is not clear, recent literature suggests a functional link between aneuploidy, unfolded protein response (UPR), and a dysregulation of macrophages and T cells [40]. In this article, the authors described how the UPR links tumor aneuploidy to local immune dysregulation. They showed that aneuploidy is a sole trigger of the UPR in cancer cells and imparts immunosuppressive and pro-tumorigenic functions to bone marrow-derived macrophages and dendritic cells, and indirectly impairs the function of T cells. A mechanistic link between aneuploidy and the UPR in cancer cells was sought using reversine, a small molecule known to induce aneuploidy through inhibition of the mitotic spindle. The authors suggested that aneuploidy directly affects the UPR in tumor cells, mainly targeting the PERK pathway and transcellular tumor-infiltrating macrophages via IRE1α and RIDD in the tumor. 

Further characterization of immune cells in the non-tumor areas of the lung allowed us to detect a significant reduction in cytotoxic CD8 T cells in the aneuploid group, and similar results were also found in patients’ lung tissues classified as high CIN. 

A different story opens up with NK cells in our model of LC. It was recently described that cells with highly aberrant karyotypes characterized by complex patterns of aneuploidies cease to divide, undergo senescence, and produce pro-inflammatory signals that lead to their elimination by NK cells in vitro [41]. In our model, there was an increase in the number of NK cells not only in the non-tumor areas but also inside the lung tumors of Alk+Mad2 mice. Moreover, a significant increase in NK cells was detected at early time points in mice that only expressed Mad2. These results tempt us to speculate that aneuploidy might be a driving force of NK cell recruitment to the lung tissue in both tumors and non-tumoral parts. Interestingly, despite this increase in NK cells, the level of IFN-γ was significantly lower in the aneuploid-bearing mice, suggesting that NK, cells being one of the main sources of IFN-γ, were no longer functioning properly as innate immune cells. Although it was recently reported that NK cells in the tumor microenvironment can convert into intILC1 and ILC1 and lose the ability to control local tumor growth and metastasis [33], we did not find any evidence of such conversion of NK cells in our mouse model. Another possible scenario for the drastic reduction in IFN-γ in the aneuploid group could relate to the diminished level of cytotoxic T cells and increased level of T cells’ exhaustion signature, which we found in LUAD patients with high CIN scores. 

Collectively, these findings indicate a crucial role of aneuploidy in shaping the tumor’s immune landscape, which might play an active role in the awakening of dormant cancer cells and tumor initiation and, as a result, higher tumor burden with more likely poor prognosis. 

## 5. Conclusions

Our results showed that aneuploid epithelial tissue contributes to increased lung tumor burden in *Eml4-Alk* mice. This higher tumor burden is not the result of increased proliferation and tumor progression, but it is rather a consequence of the immunosuppressive environment generated by aneuploid cells. We observed an increased number of TAMs, neutrophils, and AMs, a low level of IFN-y, enhanced infiltration of CD4^+^ T cells, and a reduction in CD8^+^ cytotoxic T cells in non-tumor aneuploid areas of the lung implicating CIN in the development of an immunosuppressive environment. A similar picture was also observed in human LUAD, where patients with high CIN presented a higher level of T cell exhaustion signature but a lower level of cytotoxic T cell and M1 macrophage signature. These data allowed us to speculate that an immunosuppressive microenvironment is not only playing a role during tumorigenesis but also in tumor initiation, where cancer cells are not eliminated by an active immune system.

## Figures and Tables

**Figure 1 cancers-13-06027-f001:**
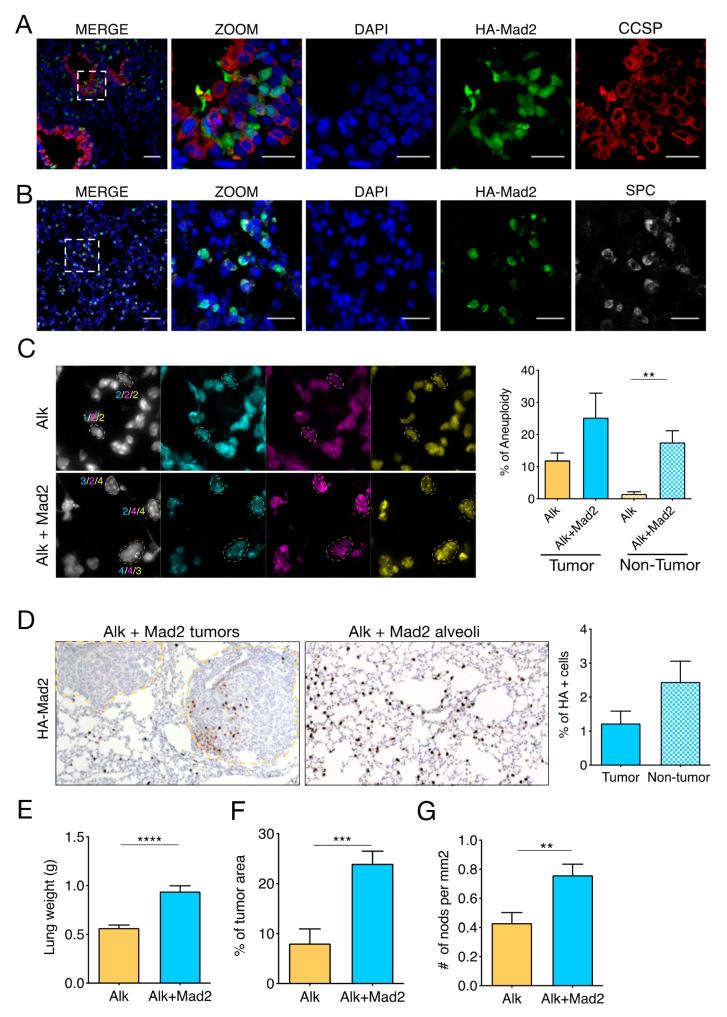
*Mad2* overexpression results in increased tumor burden in *Eml4-Alk* mice. (**A**,**B**) Immunofluorescence of HA-Mad2 and CCSP (**A**) or HA-Mad2 and SPC (**B**) showing that HA-Mad2 was expressed in Club cells and AT2 cells in the distal lung. The antibodies used are indicated. Scale bars: 60 μm. (**C**) Representative FISH images of Alk and Alk+Mad2 non-tumor areas of the lung showing euploid cells (white circle) and aneuploid cells (yellow circle). DNA: white, chromosome 12 probe (blue), 16 (magenta), and 17 (yellow). (Magnification ×63) and percentage of aneuploidy in tumors and non-tumor areas (*n* = 4 mice). ** *p* < 0.05, Two-tailed *t*-test. (**D**) Representative images of HA-Mad2 expression in tumors (left panel-yellow) and non-tumor areas (right panel) and percentage of HA-positive cells in tumors and non-tumor areas of Alk+Mad2 mice. (**E**) Total lung weight in Alk and Alk+Mad2 mice. **** *p* < 0.0001. (**F**) % of tumor area/total lung area on H&E-stained cross-sections of Alk and Alk+Mad2 groups. *** *p* < 0.05. (**G**) Number of nodules per mm^2^ in Alk and Alk+Mad2 groups. ** *p* < 0.05. For (**E**–**G**), *n* = 10 mice.

**Figure 2 cancers-13-06027-f002:**
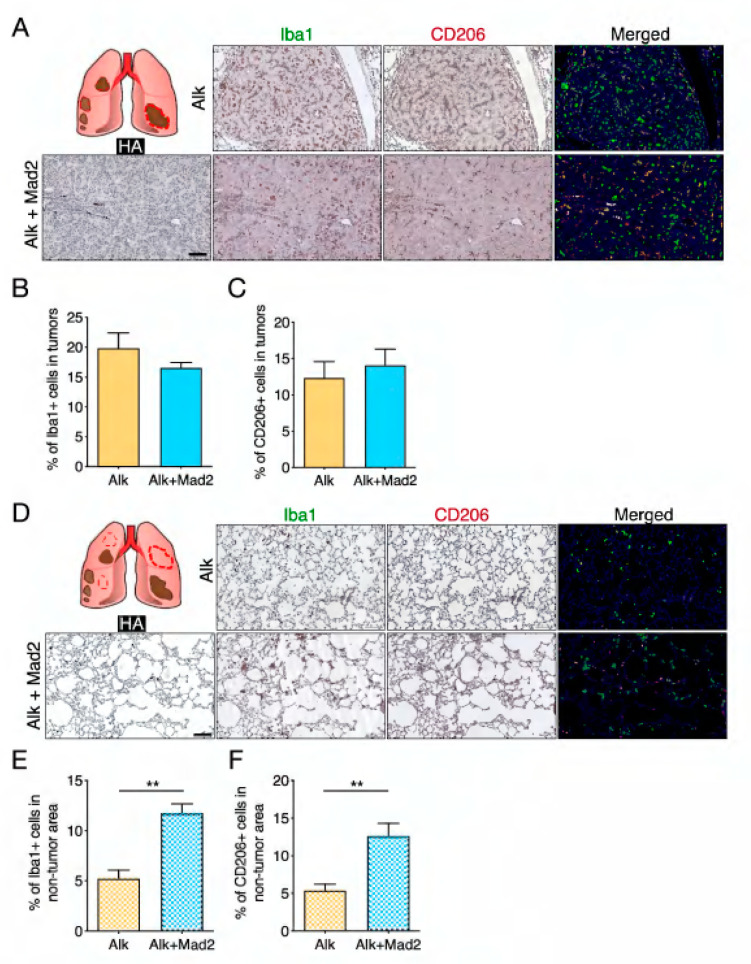
Macrophage content in lung tissues of Alk and Alk+Mad2 mice. (**A**) Representative images of multiplex immunohistochemical staining with antibodies for HA, Iba1, CD206, and their combinations (Hematoxylin-blue, HA-grey, Iba1-green, CD206-red) of tumors in Alk and Alk+Mad2 groups. Scale bar: 200 μm (**B**) Percentage of Iba1 positive cells in tumors (*n* = 4 mice). (**C**) Percentage of CD206 positive cells in tumors (*n* = 8 mice). (**D**) Representative images of multiplex immunohistochemical staining with antibodies for HA, Iba1, CD206, and their combinations (Hematoxylin-blue, HA-grey, Iba1-green, CD206-red) of non-tumor areas in Alk and Alk+Mad2 groups. Scale bar: 200 μm. (**E**) Percentage of Iba1 positive cells in non-tumor areas (*n* = 4 mice). (**F**) Percentage of CD206 positive cells in non-tumor areas (*n* = 8 mice) ** *p* < 0.05, Two-tailed *t*-test.

**Figure 3 cancers-13-06027-f003:**
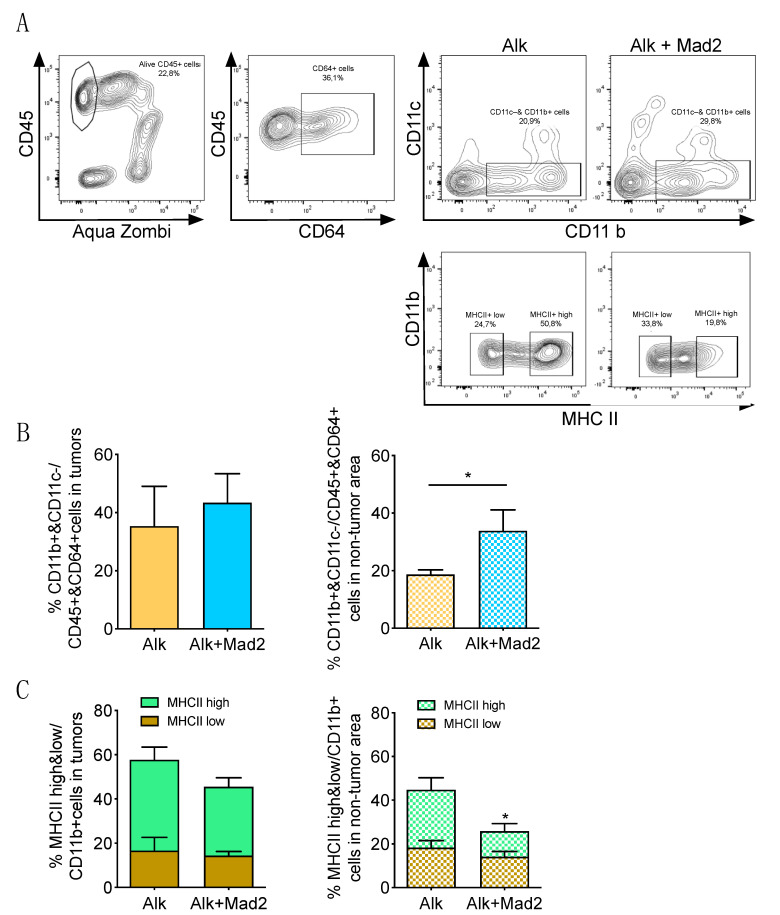
Qualitative characteristics of lung macrophages of Alk and Alk+Mad2 mice (**A**) Lung tumors and non-tumor areas were processed and stained for CD45, CD64, CD11c, and CD11b. Representative flow plots show alive CD45^+^ cells out of all lymphocytes after exclusion of debris and doublets; CD64^+^ cells in the alive CD45^+^ cells; CD11b^+^ interstitial macrophages from the CD64^+^CD45^+^AquaZombie– cells (up) and MHCII^high^ and MHCII^low^ subpopulation of CD11b^+^ interstitial macrophages (down). (**B**) Percentage of interstitial macrophages gated as CD11b^+^ and CD11c^−^ population from CD45^+^ and CD64^+^ population in tumors (Alk; *n* = 4, Alk+Mad2; *n* = 7) and in non-tumor area (Alk; *n* = 10, Alk+Mad2; *n* = 10). (**C**) Percentage of MHCII^high^ and MHCII^low^ population in CD11b^+^ and CD11c^−^ cells in tumors (Alk; *n* = 4, Alk+Mad2; *n* = 7) and non-tumor areas (Alk; *n* = 10 Alk+Mad2; *n* = 10). * *p* < 0.05; Two-tailed *t*-test.

**Figure 4 cancers-13-06027-f004:**
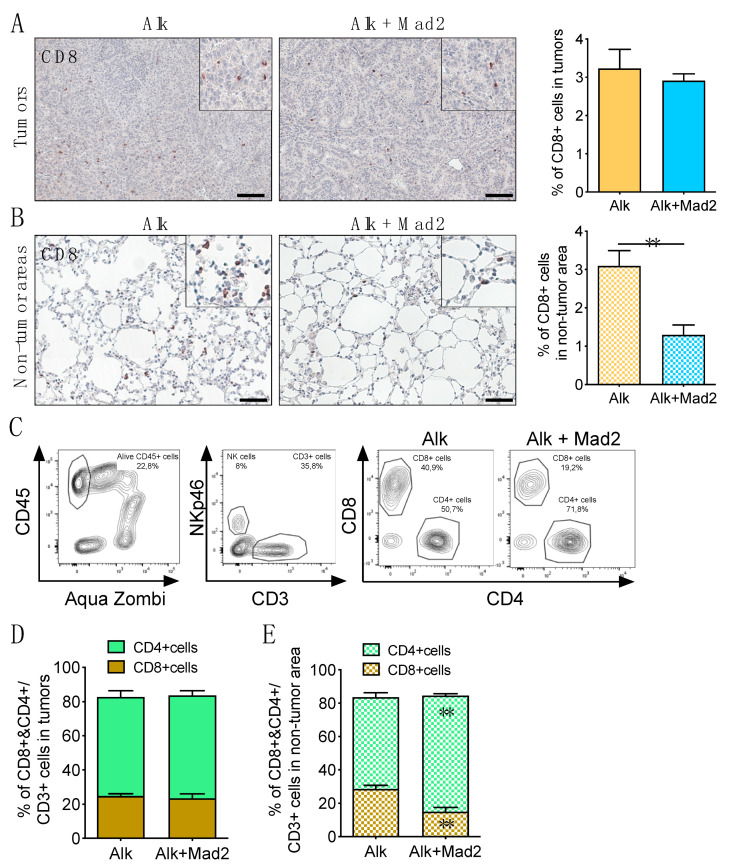
T cells and NK cells in Alk and Alk+Mad2 groups. (**A**) Representative images of CD8 T-cell infiltration in lung tumors and percentage of CD8^+^ cells in Alk and Alk+Mad2 mice (*n* = 4). Scale bar:100 μm. (**B**) Representative images of CD8^+^ cell infiltration in non-tumor areas and percentage of CD8^+^ cells in Alk and Alk+Mad2 mice (*n* = 4). Scale bar: 50 μm. (**C**) Representative flow plots of lung tumor and non-tumor areas, stained for CD45, NKp46, CD3, CD8, and CD4. (**D**) Percentage of CD8^+^ and CD4^+^ cells gated from CD45^+^ and CD3^+^ population in tumors (Alk; *n* = 10, Alk+Mad2; *n* = 6) and in (**E**) non-tumor areas (Alk; *n* = 10, Alk+Mad2; *n* = 6). ** *p* < 0.01; one-way ANOVA.

**Figure 5 cancers-13-06027-f005:**
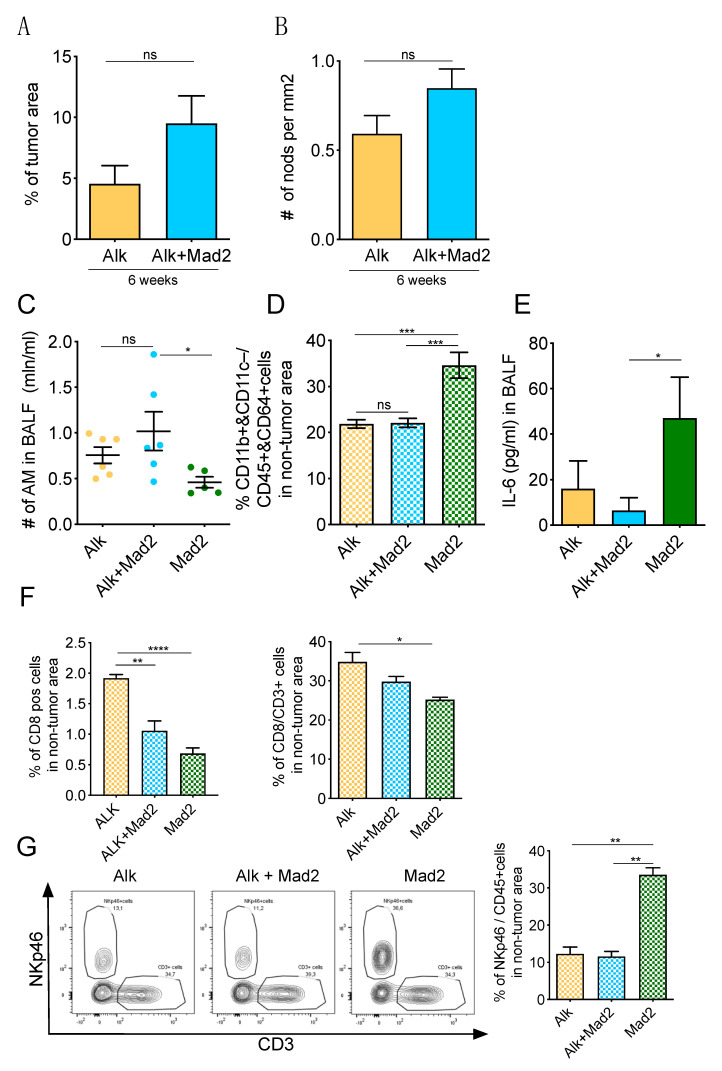
Characterization of Alk and Alk+Mad2 mice 6 weeks after tumor initiation. (**A**) Percentage of tumor area/total lung area on H&E-stained cross-sections of Alk and Alk+Mad2 mice. (**B**) Number of nodules per mm^2^ in Alk and Alk+Mad2 mice. (**C**) Number of alveolar macrophages in the bronchoalveolar lavage fluid of Alk, Alk+Mad2, and Mad2 mice. (**D**) Percentage of interstitial macrophages gated as CD11b^+^ and CD11c^−^ population from CD45^+^ and CD64^+^ population in the non-tumor area of Alk, Alk+Mad2, and Mad2 groups. (**E**) ELISA detection of IL-6 in the BALF of Alk, Alk+Mad2, and Mad2 mice. (**F**) Percentage of CD8^+^ cells in the non-tumor area of Alk, Alk+Mad2, and Mad2 lung tissues (immunohistochemical analyses, left panel) and percentage of CD8^+^ cells gated from CD3^+^ population in the non-tumor area (FACS analyses, right panel). (**G**) Representative flow plots of lung non-tumor areas, stained for CD45, NKp46, CD3, and percent of NKp46 cells from CD45^+^ cells in Alk, Alk+Mad2, and Mad2 mice. Two-tailed *t*-test in (**A**,**B**) and one-way ANOVA test for (**C**–**E**,**G**). Kruskal–Wallis, ANOVA in (**F**). In (**A**,**B**) Alk; *n* = 9 and Alk+Mad2; *n* = 9 mice. In (**C**–**G**) Alk; *n* = 6, Alk+Mad2; *n* = 6, Mad2; *n* = 5. ns, not significant; * *p* < 0.05; ** *p* < 0.01; *** *p* < 0.001; **** *p* < 0.0001.

**Figure 6 cancers-13-06027-f006:**
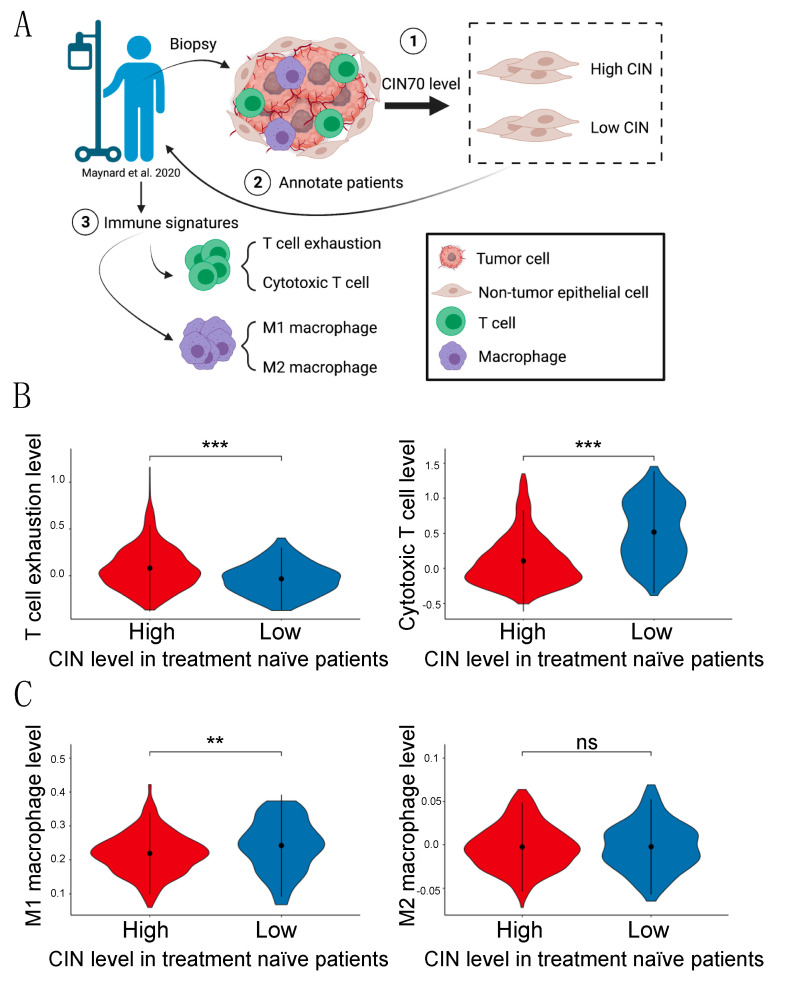
Different immune signatures in high and low CIN patients. (**A**) Schematic showing the experimental design. (**B**) Levels of T cell exhaustion and cytotoxic T cells in samples collected before treatment (treatment naïve) in high and low CIN patients. (**C**) M1 and M2 macrophages cells in samples collected before treatment (treatment naïve) in high and low CIN patients. The Y-axis shows the scores of each signature, while the dot in each graph represents the mean value and the lines connecting the dot represent the standard deviations. ns, not significant; ** *p* < 0.01; *** *p* < 0.001 Two-tailed *t*-test.

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
