# Peer review of "Mad2 Induced Aneuploidy Contributes to Eml4-Alk Driven Lung Cancer by Generating an Immunosuppressive Environment"

_cancers, 2021, doi:10.3390/cancers13236027_

Round 1

Reviewer 1 Report

Alikhanyan et al. address an important question in the field of cancer research, trying to gain a better understanding of the relationship between aneuploidy and the immune response in tumors. They specifically examined the role of aneuploidy in the tumor microenvironment on the activation of immune cells. To generate tumors, they used a previously developed sophisticated Cas9 approach to induce Eml4-Alk driven lung cancer. Then, they induced epithelial-specific aneuploidy using a KH2-Mad2 and CCSP-rtTA system. Their results show that the non-tumor regions adjacent to the Eml4-Alk driven tumors have different immunological properties when Mad2 is expressed, specifically showing changes in macrophage, T-cell, and NK cell numbers. From these data they conclude that the presence of aneuploidy in the tumor microenvironment alters the immunological landscape and provide relevant evidence from human patients. Overall, the study provides a comprehensive examination of the immune landscape in the described conditions. However, there are several caveats to this study that need to be addressed, as described below.

The title suggests two main ideas: 1) that aneuploidy is the reason for the immunosuppressive environment, and 2) that there is, indeed, an immunosuppressed environment under the conditions tested. However, the work does not address the possibility that the effects observed are Mad2 specific, and therefore are unrelated to aneuploidy. It would be optimal to provide some evidence that other drivers of aneuploidy generate similar immunological landscapes. Also, to claim that the environment is indeed immunosuppressive would require much more convincing, and this reviewer feels this is outside the scope of the current effort. I would recommend changing the current title and the claims in the text to say they found an “immunosuppressive signature, or profile”, emphasizing that to test the immunosuppressive quality of the environment would require additional experiments.

Main points:

  1. Is the effect due to aneuploidy or Mad2 expression? Ideally, another aneuploidy model should be examined. Alternatively, an experiment where Mad2 is induced and then doxycycline withdrawn (and Mad2 levels are reduced) could help! Would the immune landscape change? A careful examination of Mad2 levels and aneuploidy should accompany such an attempt – but this could be very telling.
  2. The definition of “non-tumor area” is lacking. Could it be that in the Alk-Mad2 tumors there are many “microtumors” (as Mad2 induction increases tumor initiation), and thus the so-called “non-tumor area” in these mice actually contains tumor cells? Colocalization of Eml4-Alk inversion and Mad2 (or aneuploidy) could shed light on this.
  3. The Mad2 mouse background in unclear, also from the methods. Please provide a precise explanation of these mice, where is Mad2 expressed, and where does it induce aneuploidy. A more critical examination of aneuploidy in the different cell types in these mice is required.
  4. Profiling of the immune landscape in the paper is confusing. It is hard to understand when an effect is observed, and if it supports the idea of immunesuppression. The authors should clarify this much better. Also, to support the idea that the immune cells recruited to the aneuploidy environment are different, it would be ideal to sort different immune populations and analyze their transcriptome – single cell RNA sequencing is optimal, but bulk sequencing could provide much support to the idea that in such cases the cells are indeed different. This way, we would not need to rely solely on marker expression.
  5. What is the difference between aneuploid cells in the tumor microenvironment and aneuploid cells unrelated to tumors? Would it be that in both cases immunesuppressive cells would be recruited to the aneuploid region? What would be the level of AM, IL6, NK, T-cells in aneuploid lungs without tumors?

Additional comments:

Line 11: “aneuploid cells can be recognized and eliminated by immune cells” – I am not sure this was directly shown in the past and would tone this argument a little bit.

Lines 50-52: “Evidence obtained in nonmalignant cells and cancer models indicates that chromosome instability can favor the development and selection of malignant clones, driving progression to a very aggressive phenotype” – There are many citations that the authors should consider adding to support this sentence.

Line 67: “Nevertheless the impact that an aneuploid tumor environment might have on tumor progression has not yet been studied”. What about the possibility that the tumor itself could induce aneuploidy in the environment? How would the authors suggest that aneuploid environments form? This is actually an important point – is there a reason to believe that aneuploidy is a feature of the tumor microenvironment that is independent of the tumor? This is required to substantiate our interest in the question examined in the paper.

Lines 79-83: “Moreover, our findings reveal that an aneuploidy environment increases the number of alveolar macrophages (AMs) as well as tumor associated macrophages (TAMs) and neutrophils, while the level of cytotoxic CD8+ T cells and IFN-γ in lung tissue decreases, thus generating an immunosuppressive environment, leading to a higher tumor burden.” – There is no direct cause and effect between the levels of CD8+ T cells and IFNgamma, and a higher tumor burden. This is correlative.

Figure 1C: FISH quality is lacking. Better images are required, and this experiment could be complemented with single-cell DNA sequencing to provide an estimation of aneuploidy in a less biased (or subjective) way. Also, the aneuploidy difference between Alk and Alk-Mad2 tumors actually looks like it could be significant, perhaps with more repeats?

Line 291: “later” should be “latter”.

Figure 2D: Alk is more dense than Alk-Mad2 – are the regions examined comparable?

Figure 2C,F: wouldn’t a pairwise comparison of non-tumor and tumor areas from the same mouse be appropriate?

Lines 299-302: too technical – would be better in the methods section.

Figure 3A: the percentages are not clear.

Figure 3: How do you know if the cells taken for the FACS analysis are from the tumor or non-tumor areas.

Lines 350-351: “Altogether, the increase of CD4/CD8 ratio, might result in increased tumorigenesis.” – How would the authors support this claim?

Figure 4A/B: could you provide zoom insets – it is difficult to see an differences.

Figure 4C: percent is not clear.

Sup 3G: The difference in % of Ly6G needs more context: is this considered a large/small increase? Some positive control or reference that tells us the meaning is required.

Figure 5 is confusing: now, Alk and Alk-Mad2 are similar, but the Mad2 alone is different. How could this be?

Figure 6: Although the difference identified appear mild, I appreciate the difficulty in examining this idea in data from human patients. A better explanation of the CIN70 score is needed. Although not essential, this figure could benefit from additional analyses (maybe comparing MIN+ and CIN+ colon cancer patients?).

Reviewer 2 Report

Alikhanyan, Chen et al in this manuscript provide evidence that Mad2 driven aneuploidy in the context of a Eml4-Alk driven mouse model of NSCLC contributes to increased tumor intiation through immunosuppression mechanism. Overall, the manuscript is well-written with clear data and interpretations and supported by an external dataset. 

Some significant issues to note:

  1. Although the others clearly show that at 16 weeks, Mad2 is associated with more tumor nodules and show increased interstitial macrophages they do not provide a causal link between these two observation. For example, the authors cannot claim that immunosuppression environment is necessarily causing the increased tumor initiation. Further experiments to prove this causal link should be undertaken or language in the manuscript should be properly adjusted.
  2. The inclusion of generally negative data at 6 weeks is appreciated. However, the authors should more explicitly discuss why they think mad2 overexpession leads to immunosuppression changes mainly through macrophages at 16 weeks vs. 6 weeks. 
  3. For the single cell data shown in Fig. 6 the authors should given more details of patients in this cohort such as how EML4-ALK driver as this is the focus of their mouse model and the manuscript. They also need to clarify if the analysis shown are all from non-tumor cells? 
  4. Flow gating strategy should be provide in the supplementary
  5. Authors should comment on why they are using Eml4-Alk model to study this and whether their results could apply to other types of NSCLC? Such as EGFR mutant lung cancer? 
  6. Some comment should be made for the potentials mechanisms by which aneuploidy may specifically alter macrophage content in normal tissue surrounding tumors. 
  7. How do the authors think Alk specific inhibitors may impact their results? Such experiments may be out of the scope of this paper but it should be mentioned and discussed. 
  8. Did the authors look at other immunosuppression markers such as immune checkpoints, PD-L1, LAG3, TIGIT or others in their model?
  9. The authors should show some orthogonal method in addition to confirm their main finding of macrophage induction, such as by IHC

Round 2

Reviewer 2 Report

The authors have made sufficient revisions to my satisfaction.